# FUSION TOKEN: ENHANCING COMPRESSION AND EFFICIENCY IN LANGUAGE MODEL TOKENIZATION

## ABSTRACT

In the realm of language models, data encoding is pivotal, influencing efficiency and effectiveness of model training. Byte Pair Encoding (BPE) is a well-established subword tokenization technique that balances computational efficiency and linguistic expressiveness by merging frequent byte or character pairs. As language model training requires substantial computational resources, we propose Fusion Token, a method that substantially enhances the conventional Byte Pair Encoding (BPE) approach in data encoding for language models. Fusion Token employs a more aggressive computational strategy compared to BPE, expanding the token groups from bi-grams to 10-grams. Remarkably, with the addition of 1024 tokens to the vocabulary, the compression rate significantly surpasses that of a regular BPE tokenizer with a vocabulary of one million. Overall, the Fusion Token method leads to noticeable performance improvements due to an increased data scope per compute unit. Additionally, higher compression results in faster inference times due to fewer tokens per given string. By devoting more compute resources to the tokenizer building process, Fusion Token maximizes the potential of language models as efficient data compression engines, enabling more effective language modeling systems.

## 1 INTRODUCTION

Language models play a transformative role in numerous areas of artificial intelligence, including natural language understanding, information retrieval, text summarization, sentiment analysis, machine translation, and dialogue systems. Their inherent versatility has placed large language models (LLMs) at the forefront of current technological advancements, driving state-of-the-art results in a broad range of tasks (Brown et al., 2020; OpenAI, 2023a; Chowdhery et al., 2022; Touvron et al., 2023; Chen et al., 2021; Hoffmann et al., 2022; Li et al., 2022; Microsoft; Amazon; Nijkamp et al., 2023).

Data encoding into tokens is a critical component in shaping the efficiency and effectiveness of language models, serving to transform data into a digestible format for model training. Byte Pair Encoding (BPE), a renowned subword tokenization method, has been a cornerstone in this task. BPE operates on a principle of computational balance, methodically constructing a vocabulary of tokens by identifying and merging the most frequent pair of bytes or characters in the data. This strategy aims to strike a balance between computational efficiency and linguistic expressiveness.

However, the recent shift in the landscape of language model training, which involves dedicating increasingly larger computational resources (OpenAI, 2023a; Chowdhery et al., 2022), opens up the possibility of investing more resources into tokenization to enhance its effectiveness. This need becomes more pronounced considering the substantial computational resources dedicated to language model training to reduce cross entropy.

The study introduces Fusion Token, a novel method enhancing the traditional Byte Pair Encoding (BPE) process used for data tokenization. Fusion Token allows for merging general n-grams, as opposed to just bi-grams, leading to a notable improvement in compression rates by prioritizing tokens with higher bytes during merging. This enhancement to the BPE method significantly refines the vocabulary construction, achieving a compression rate that outperforms that of a BPE vocabulary of one million with just an additional 1K tokens to a 51K vocabulary.

Fusion Token also advances language model performance across various benchmarks including code generation. We hypothesize that these improvements are attributed to its superior compression rate, allowing for the processing of more information during model training given the same token budget. Moreover, Fusion Token delivers lower generation latency, enabling faster inference times. This efficiency in tokenization, reducing the number of tokens required for any given text or code, is beneficial for real-time applications demanding rapid and more efficient language modeling.

## 2 TOKENIZATION AND LANGUAGE MODELING

### 2.1 BYTE-PAIR-ENCODING AND BYTE-LEVEL BPE

Byte Pair Encoding (BPE), initially developed as a data compression technique, has found widespread use in natural language processing for tokenization, especially in training language models (Sennrich et al., 2016; Radford et al., 2018). BPE efficiently encodes data, not limited to text, by replacing frequently occurring byte pairs with a single unused byte, leading to an optimal balance between computational efficiency and linguistic expressiveness.

For universal language model applicability, tokenizers must be capable of encoding all text. Therefore, many modern BPE implementations operate at the byte level, resulting in a byte-level BPE, using the 256 unique bytes as the fundamental units as a base vocabulary which can represent arbitrary Unicode characters (OpenAI, 2023b). This byte-level approach guarantees lossless encoding of any Unicode string, enhancing the model's robustness and versatility. An example is the character "é", represented by two bytes (C3 and A9) in UTF-8. Initially, a byte-level tokenizer treats "é" as two separate tokens. However, if "é" is common in the text, the iterative BPE process might merge these tokens into one, representing "é". This is in contrast with the character-level BPE where the space of all possible unicode characters is too large ($\approx 150000$ and growing).

Pre-tokenization, an initial processing step, often complements the main tokenization process before the BPE process. This step partitions the input text into manageable "pre-tokens" using basic delineators like whitespace or punctuation based on a pre-designed regular expression. These pre-tokens are then further tokenized by the main process, such as BPE, into subword tokens. The primary advantage of pre-tokenization is the initial breakdown of text into linguistically meaningful units to facilitate the main tokenizer building process, including the usage of schema such as space-prefix.[1] During the tokenizer building stage, the BPE algorithm performs the bi-gram merging entirely on the pre-token space, which helps control the combinatorial complexity and computation.

### 2.2 LANGUAGE MODEL AND BITS PER BYTE

Language model efficiency is closely tied to concepts like cross entropy and bits per word (BPW) or byte (BPB). Cross entropy measures the average bits necessary to encode the true word distribution (P) using the model's predicted distribution (Q), essentially representing BPW for the language model's encoding.

$$\text{Bits per word (BPW)} = H(P, Q) = -\sum_i P(i) \log Q(i) = \text{LM cross entropy}$$

The bits per bytes (BPB) is $H(P, Q)/E[\ell]$ where $E[\ell]$ is the average token length in bytes. In essence, the average number of bits required to represent a word is equal to the average number of bits required to represent a byte multiplied by the average length of a token in bytes $E[\ell]$.

The relevance of tokenization comes into play with the concept of bits per byte (BPB), a metric that ties the tokenization strategy with the information efficiency of the language model. BPB effectively measures the average bits required per byte, indicating the efficiency of the language model and tokenization strategy in representing the information. A more effective method will result in a lower BPB, indicating a more compact and efficient representation of the text data.

---

[1]Space-prefix refers to the common notation of grouping a single white space before a non-whitespace text during the pre-tokenization. BPE training itself would not yield such consistent notation without such pre-defined schema.

## 2.3 BITS PER BYTES UPPER BOUND

The relationship between bits per word (BPW) and bits per bytes (BPB) may underline the significance of optimizing tokenization to achieve efficient text encoding in language models. Reducing the average token length through optimization can potentially lead to a more compact representation of the text corpus, which in turn enhances language model performance.

We can explore a few cases to better understand the upper bound of entropy and the effects of tokenization on compression. First, let us consider a case where the vocabulary size V is equal to the number of unique bytes, i.e., $V = 256$. In this scenario, each byte corresponds to one token in the vocabulary, and the token length is always one byte. The upper bound of the entropy is when the distribution $Q(i)$ assigns equal probability to each token, i.e., $Q(i) = 1/V$. That is,

$$BPB = \frac{H(P,Q)}{E(\ell)} \leq logV \tag{1}$$

In this case, the expected token length, $E[\ell]$, is also equal to 1. The bits per bytes is $BPB \leq -\log_2(1/V)/E(\ell) = \log 256$.

The second case involves an inefficient encoding where the vocabulary includes all possible byte combinations of length $\ell$, i.e., $V = 256^\ell$. In this setting, the expected token length is $E[\ell] = \ell$. This results in the same upper bound of BPB as the previous case. In other words, no compression is achieved despite the increase in vocabulary size.

These two cases illustrate that (1) merely increasing the vocabulary size does not guarantee efficient compression. The upper bound of cross entropy remains the same in both cases, highlighting the importance of building the vocabulary via optimizing tokenization in order to achieve effective compression. (2) that the scale of bits per byte metric is less susceptible to the vocabulary size, as opposed to the bits per word metric which depends strongly on V and hence cannot be used to indicated the compression.

In general, we can express the vocabulary $V = r^{E(\ell)}$ where $r = V^{\frac{1}{E(\ell)}}$ can be viewed as a tokenizer compression rate or the number of possible bytes at each position. In this case,

$$BPB \leq \frac{\log V}{E[\ell]} = \log r \tag{2}$$

That is, the bits per bytes's upper bound is log the number of bits to represent at each token in the vocabulary.

## 2.4 BPB AND LEARNING EFFICIENCY

We aim to primarily increase compression rate $E(\ell)$ without much increase in vocabulary size, which would serve the reduce such lower bound $\log r$, hence improving the compression by a tighter compression bound. We hypothesize and demonstrate empirically that a tokenizer that leads to lower compression will result in lower bits per byte, given sufficient model capacity to learn more complex patterns due to increased compression. If that is the case, then the lower BPB can also result in a more capable language model based on downstream performance due to the artifact of being able to train models with more passes over the data given the same total token budget during training.

## 3 FUSION TOKEN

We outline the Fusion Token in Algorithm 1. The Fusion process starts from taking an existing BPE vocabulary and iteratively group up to n-grams of tokens, where we use $n_{max} = 10$ in practice. To optimize the compression rate, at each step we select the token that has highest probability of occurrences. While theoretically we can directly prioritize the expected bytes (probability of each n-gram times the number of bytes), by sorting the product of probability and the number of bytes instead of using the probability alone, we find that it does not matter much since the scale of the variability of observed probability is orders of magnitude higher than that of the number of bytes.

---

**Algorithm 1** Token Fusion

---

**Input:** Existing vocabulary $V$, additional vocabulary size $\Delta V$, dataset of pretokens $D$
**Output:** New vocabulary $V'$ after token merging
$V' \leftarrow V$
**for** $i = 1$ **to** $\Delta V$ **do**
  $max\_value \leftarrow 0$
  **for** each unique $n$-gram $d_n \in D$ **do**
    Compute the length adjusted frequency
    $value \leftarrow (n-1)\mathbb{P}(d_n|D)$
    **if** value $>$ max\_value **then**
      $t_{\text{new}} \leftarrow$ grouped tokens from $d_n$
      max\_value $\leftarrow$ value
    **end if**
  **end for**
  Add $t_{\text{new}}$ to $V'$
**end for**

---

Table 1: Bytes per token for text and programming languages for the SentencePiece BPE tokenizer with 51K vocabulary, the 51K BPE + 1K Fusion Token, and the 1M BPE. The additional 1K tokens ($\approx 2\%$ over 51K original vocabulary) results in much improvement on the compression rate on all data domains. $\% \uparrow$ column represents compression rate improvement over the its baseline tokenizer. Code row represents the weighted average of all non-text columns based on code availability in public sources.

|        | 51K  | 51K+1K Fusion | ($\% \uparrow$) | 1M   |
|--------|------|---------------|-----------------|------|
| Text   | 3.25 | 3.33          | 2.57            | 3.70 |
| C      | 2.24 | 2.44          | 9.02            | 2.34 |
| C++    | 2.37 | 2.58          | 8.54            | 2.49 |
| C#     | 3.02 | 3.31          | 9.62            | 3.15 |
| Go     | 2.33 | 2.50          | 7.32            | 2.42 |
| Java   | 2.84 | 3.08          | 8.47            | 3.00 |
| JS     | 2.62 | 2.92          | 11.3            | 2.71 |
| Kotlin | 2.97 | 3.22          | 8.14            | 3.15 |
| PHP    | 2.60 | 2.89          | 11.4            | 2.67 |
| Py     | 2.68 | 2.87          | 7.27            | 2.81 |
| Ruby   | 2.56 | 2.78          | 8.76            | 2.65 |
| Rust   | 2.57 | 2.79          | 8.74            | 2.64 |
| Scala  | 2.69 | 2.94          | 9.27            | 2.84 |
| Shell  | 2.29 | 2.53          | 10.6            | 2.45 |
| SQL    | 2.52 | 2.72          | 8.11            | 2.68 |
| Code   | 2.66 | 2.91          | 9.43            | 2.78 |

After selecting the highest probability n-gram, we add it to the vocabulary and perform the same process iteratively until we reach the desired number of additional fused tokens. Note that the traditional BPE can be seen as a special bigram case of Fusion Token.

During inference or tokenization stage, fusion tokens are then added to the vocabulary as special tokens that takes precedent over regular tokens. That is, whenever the fusion tokens are present in each string, they are prioritized to be used for the first stage of tokenization. Then the rest of the process follows normal bigram BPE.

## 4 EXPERIMENTS

Throughout the experiments, we use text and code data for tokenizer training (additional details in Section 4.2). We primarily use a modified SentencePiece Byte Pair Encoding (BPE) setting where we operate on the byte-level representation and also ensure a lossless tokenizer by including all base bytes to the vocabulary. To ensure applicability, we also test fusion token on another BPE

Table 2: Code data and the corresponding sizes

| Data | C | C++ | C# | Golang | Java | JS | Kotlin | PHP | Python | Ruby | Rust | Scala | Shell | SQL | TS |
|------|------|------|------|--------|------|------|--------|------|--------|------|------|-------|-------|------|------|
| Size | 8.1G | 15G | 22G | 5G | 58G | 45G | 1.1G | 26G | 37G | 3.1G | 1.9G | 0.93G | 1.3G | 2.2G | 11G |

Table 3: SentencePiece Settings

| model type | bpe |
|------------|-----|
| character coverage | 0.9999 |
| input sentence size | 10,000 |
| max sentence length | 1,000,000,000 |
| max sentencepiece length | 8,384 |
| add dummy prefix | True |
| remove extra whitespaces | True |
| allow whitespace only pieces | True |
| split by whitespace | True |

implementation based on HuggingFace BPE trainer (Wolf et al., 2019). Details of the data split can be found in Table 3.

## 4.1 SENTENCEPIECE TOKENIZER TRAINING

In order to train the baseline tokenizer used in our experiments we used Sentencepiece-BPE with the settings described in 3. This model covers 14 languages for generality and is trained up to 51200 vocab length for the standard tokenizer and 1,024,000 tokens in the large vocab case. Both of these models were trained using standard sentencepiece libraries found at https://github.com/google/sentencepiece.

## 4.2 TOKENIZER TRAINING DATA

We use permissively licensed data consisting of code in multiple programming languages with details outlined in Table 2. Our code data was collected from a number of online repositories including open source libraries as well as permissive code taken from https://github.com. The data was categorized according to its extension for the purpose of attributing language. Finally our text data is a subset of the Pile Gao et al. (2021).

## 4.3 COMPRESSION RATE AND FUSION TOKEN

Table 1 outlines the results where we perform Fusion Token as outlined in Algorithm 1 with 1024 additional vocabulary size on top of a 51200 vocabulary BPE tokenizer. The row 51K setting denotes the SentencePiece BPE tokenizer with 51K vocabulary. The +1K setting denotes the 51K the fusion token with additional 1024 tokens. Note here that we observe improved compression rate on text and code data in all categories (positive % indicates the increased bytes per token with fusion token). Throughout this section, we perform an analysis on the behavior of Fusion Token as well as the effects on language model performance. Below are our observations.

### 4.3.1 FUSION TOKEN OFFERS HIGH COMPRESSION FOR MODEST VOCABULARY SIZE

In Figure 1a illustrates the compression achieved from Fusion Token versus that from increasing vocabulary. In this experiment, we use a vocabulary size up to 1 million tokens. We can see that the bytes per token grows sharply up to 50K or so, after which it observes much slower growth. With the 1K fusion token on top of the 51K vocabulary, the bytes per token increases sharply and exceeds that of the 1M vocabulary tokenizer.

We also show the compression rate (bytes per token) with respect to the number of additional fused tokens on top of the 51K BPE vocabulary in Figure 1b. We can see that the bytes of token increases rapidly on top of the base compression rate (of the 51K vocabulary). With only 1K added tokens, we capture a lot of the gain in compression, resulting in a highly compact vocabulary size.

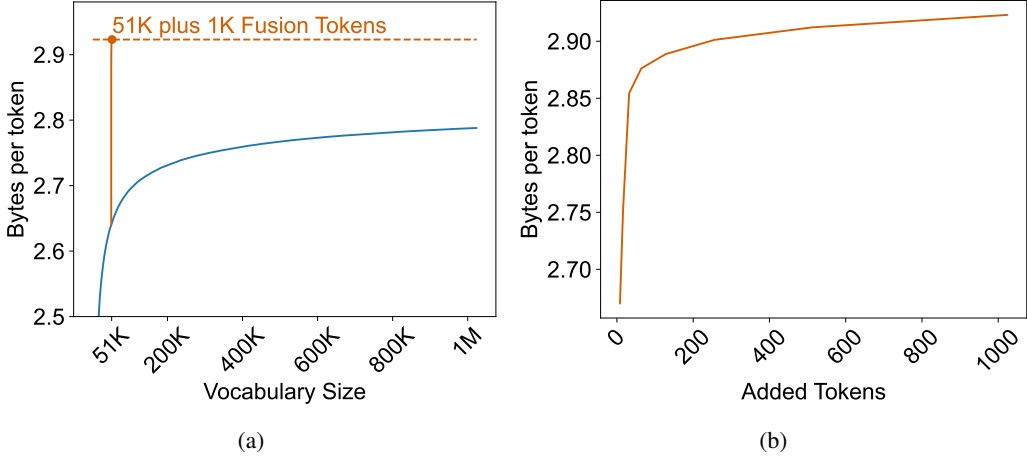

(a)                           (b)

Figure 1: This plot compares the bytes-per-token efficiency of BPE tokenizers across different vocabulary sizes, extending up to 1 million. Fusion Token's methodology, when contrasted with regular BPE, demonstrates a substantial improvement in compression rate, effectively outperforming regular BPE approach even at 1 million vocabulary size. (b) Compression rate versus the number of additional fused tokens on top of the 51K BPE tokenizer.

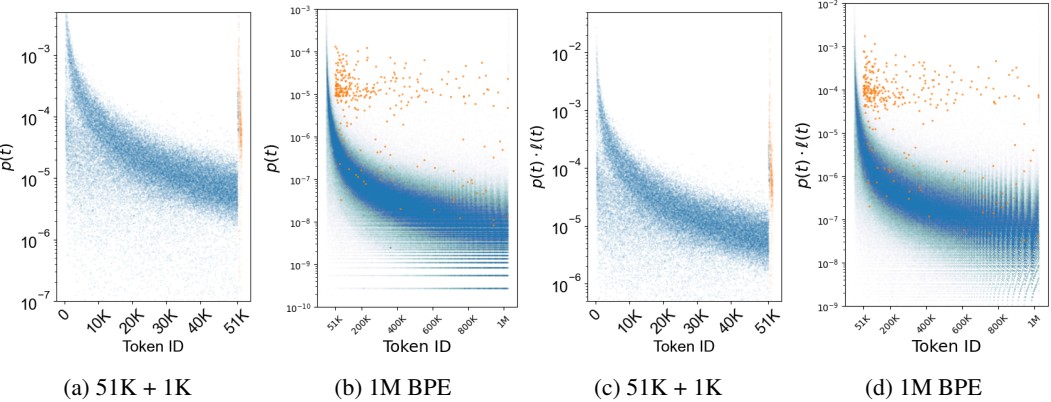

(a) 51K + 1K            (b) 1M BPE            (c) 51K + 1K            (d) 1M BPE

Figure 2: The plot illustrates the occurrence probability $p(t)$ of token $t$ appearing in the data distribution, where the orange dots indicate fusion tokens. In (a), the fusion tokens correspond to index $51k$ to $51k + 1k$ which demonstrate relatively high occurrence probabilities. In (b), the fusion tokens that align with the tokens in 1M vocabulary also have high probabilities, but not discovered during the first 51K BPE vocabulary. In (c) and (d), we show similar plots where we use show the token probability times the byte length (expected byte length), which is a metric that can directly affect compression (bytes per token). We observe that the expected byte length follow roughly the same trend as the probability, where the probability itself is the dominating factor influencing the compression of Fusion Token to be higher, since the Fusion Token probability differs from normal tokens probabilities by orders of magnitude higher. This suggests that the fusion tokens are extremely likely during regular BPE vocabulary building, but do not get incorporated due to the algorithm constraint on bi-grams.

### 4.3.2 FUSION TOKENS OCCUR FREQUENTLY BUT ARE NEGLECTED IN BPE

We find that fusion tokens can have much higher occurrence probabilities. Figure 2c illustrates the probability versus the token indices where we see that the probability varies within a few orders of magnitude in the vicinity of adjacent IDs where the overall macro tends has decreasing probabilities according to power law (Zipf's law). However, at 51K, we observe a sharp increase in such probabilities, corresponding to the fusion tokens (orange dots) which occupy the token indices

Table 4: Bits per word (cross entropy validation loss) and bits per byte of language models.

| Model | Tokenizer | BPW | Bytes/Token | BPB ↓ |
|-------|-----------|-----|-------------|-------|
| 125M | BPE 51K | 0.8233 | 2.625 | **0.314** |
| 125M | +1K Fusion | 0.9255 | 2.915 | 0.317 |
| 650M | BPE 51K | 0.6850 | 2.625 | 0.261 |
| 650M | +1K Fusion | 0.7584 | 2.915 | **0.260** |

Table 5: Pass@k results on code generation benchmarks MBXP and Multi-lingual HumanEval (JavaScript subset) for language models trained with the $51K$ BPE and $51K$ BPE $+ 1K$ Fusion Token tokenizer. The scores suggest a positive impact on downstream evaluation, potentially attributable to the more compact data representation facilitated by Fusion Token.

| Model Size | Tokenizer | MBXP-JS | | HumanEval-JS | |
|------------|-----------|---------|--------|--------------|--------|
| | | pass@1 | pass@5 | pass@1 | pass@5 |
| 125M | BPE 51K | 3.40% | 9.21% | **3.11%** | 4.35% |
| 125M | +1K Fusion | **4.37%** | **10.97%** | 2.86% | **4.97%** |
| 650M | BPE 51K | 7.10% | 15.73% | 5.96% | 9.94% |
| 650M | +1K Fusion | **8.05%** | **17.18%** | 5.96% | **10.56%** |

51K to 51K + 1024. Fusion Token method is able to incorporate such high-probability tokens which leads to overall compression improvement.

We also find that **all** of the 1K fusion tokens appear in the 1M vocabulary BPE tokenizer, demonstrated in Figure 2d where we plot the probability of token occurrences in the data. We emphasize that even though the 1M vocabulary subsumes all of the 50K BPE + 1K fusion tokens, Fusion Token enables much more efficient data encoding due to more compact representation of 51K + 1K, roughly 20 times lower.

## 4.4  EVALUATION

We test the performance of language models trained on primarily JavaScript data, tokenized using two different approaches: (1) using a BPE tokenizer with a vocabulary size of $51K$, and (2) leveraging a combination of $51K$ BPE and $1K$ Fusion Token. These evaluations are conducted across two model sizes, $125M$ and $650M$ (refer to Section 4.2 for further details on the data).

In Table 4, we show the validation loss (bits per word) as well as the bits per byte (BPB) (Section 2.2). Interestingly, we observe that while the both the validation loss and the BPB of the 125M with Fusion Token is worse than with BPE, as we scale the model size to $650M$, the BPB becomes roughly equal. If this trend continues, it is likely that the BPB for large models with Fusion Token will be better than with BPE. We leave experiments on larger model sizes for future work due to resource and time constraint.

As observed in Table 5, models trained with our Fusion Token tokenizer obtain consistently higher evaluation scores (pass@k) (Kulal et al., 2019) on code generation benchmarks such as Multi-HumanEval and MBXP (Chen et al., 2021; Austin et al., 2021; Athiwaratkun et al., 2022). This aligns with our initial hypothesis: a more compact data compression facilitates models to process higher volumes of information during training (given the same compute resources), thus potentially enhancing the overall performance of the language models.

Table 6 also indicates a significant improvement in latency (10.19%) for language models trained with 51K+1K Fusion Token during the MBXP JavaScript function completion. This latency reduction is due to the fewer number of tokens required to represent the same amount of text based on the higher compression rate. We observe that the improvement is even more pronounced that the average compression rate on JavaScript (11%), likely due to the nature of the evaluation data being more generic than arbitrary code in the wild. These enhancements underscore the effectiveness

of FusionToken, demonstrating potential for substantial performance improvements in real-world applications.

Table 6: Comparison of number of tokens and inference time for different languages for the baseline BPE and BPE + Fusion Token.

|  | Tokenizer | Python | Java | Javascript | C# | Typescript | Avg |
|---|---|---|---|---|---|---|---|
| **BPE** | # Tokens | 20.43 | 34.31 | 19.39 | 26.97 | 19.86 | 24.19 |
|  | Inf. Time (ms) | 341.23 | 529.98 | 334.59 | 521.18 | 363.42 | 418.08 |
| **Fusion** | # Tokens | 18.46 | 30.93 | 17.19 | 23.50 | 17.43 | 21.50 |
|  | Inf. Time (ms) | 313.11 | 480.15 | 302.92 | 454.93 | 326.11 | 375.44 |
| **Diff.** | ↓ # Tokens | 9.64% | 9.85% | 11.34% | 12.86% | 12.23% | 11.12% |
|  | ↓ Inf. Time (ms) | 8.24% | 9.4% | 9.46% | 12.71% | 10.26% | 10.19% |

## 5 RELATED WORK

Public implementations of byte-level BPE includes Huggingface's Tokenizers library which also provides a framework for training custom tokenizer with BPE (Inc., 2022). Other trained byte-level BPE tokenizers include GPT's tokenizer (Radford et al., 2018), BLOOM (Scao et al., 2022). SentencePiece (Kudo & Richardson, 2018) is another framework and provides ways to do tokenization with either BPE or UnigramLM.

Despite the widespread adoption of BPE across numerous language models (Radford et al., 2018; Chowdhery et al., 2022), some recent studies have started to question its efficacy. A noteworthy critique is presented by Bostrom & Durrett (2020), who conducted a comprehensive comparison between BPE and UnigramLM. The study concluded that while there were only minor differences in performance between the two methods, BPE could be considered suboptimal. Another line of work, such as TokenMonster (Alasdair, 2023), aims to achieve more optimal tokenizer compression by utilizing additional computational resources. TokenMonster employs a more complex and computationally expensive procedure to create a more compressed and efficient tokenization, contributing to enhanced performance in language models.

There are also many earlier works that focused on using ngrams as representations for subwords (Bojanowski et al., 2016) as ngrams maximizing the joint probability over the token occurrences. This work, while in a different field, notably also shows an improvement in the performance of its downstream task (perplexity in RNN based language modeling). Furthermore, qualitative analysis by the authors suggest that the clustered neighborhoods of ngrams found also adhere to their semantic meanings, suggesting that these ngrams are both compressive and semanticially representative.

## 6 DISCUSSION

The ongoing advancement in tokenization techniques represents a stride towards enhancing language model performance. By enabling a more condensed representation of data, Fusion Token contributes to valuable progression towards more efficient language models. In this paper, we use primarily use the code domain to investigate the effects of compressed tokenizer; a future direction includes a study towards more optimal tokenizer for multi-lingual natural language.

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

# A    APPENDIX

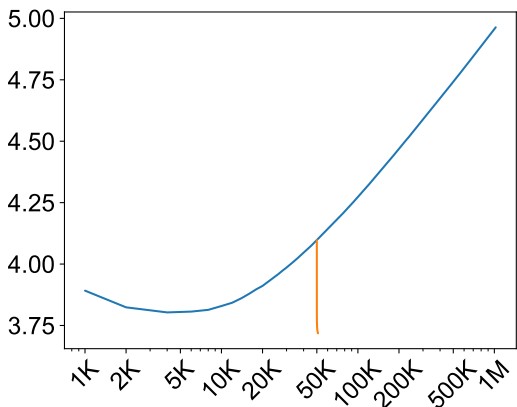

Figure 3: This chart shows $\frac{\log V}{E[\ell]}$ of the sentencepiece tokenizer as it grows to a vocabulary of 1M tokens. The orange line however represents the addition of 1K fusion tokens.

The above chart represents the upper bound on the bits per byte (BPB) as the tokenizer's vocabulary size increases. As we have discussed throughout the paper, BPB, a measure of compression, can have effects on the performance of language models in downstream tasks. We observe that the upper bound pictured in Figure 3 continues to rise with the addition of new tokens up to 1M. This is due to the growing the $\log(V)$ while the expected number of tokens per byte ($E[\ell]$) does not increase as rapidly. However with the sharp increase in compression provided by fusion tokens does improve on the bound. Notably, this bound is very loose, overstating the BPB by 10x relative to our empirical observations. Establishing tighter bounds is left to future work.

