# OpenReview forum: "Fusion Token: Enhancing Compression and Efficiency in Language Model Tokenization"
_ICLR.cc/2024/Conference — Submitted to ICLR 2024_

### Official Review · Reviewer_dMCg · 2023-10-28

**Soundness:** 3 good
**Presentation:** 2 fair
**Contribution:** 2 fair
**Rating:** 3
**Confidence:** 3

**Summary:**

This paper proposes FusionToken as an approach to improve the standard bpe tokenization and shows this approach improve tokenization efficiency: adding 1k additional vocabulary over 50K original vocabulary can achieve better tokenization efficiency (i.e., shorter sequence) than using 1M BPE vocabulary. The shorter sequences owing to the efficient tokenization can make inference faster, making language modeling efficient.

**Strengths:**

The strengths of this paper are two-fold:

- It shows promising results that adding 1k additional vocabulary can achieve better tokenization efficiency than 1M bpe tokenization in code data, demonstrating a potentially useful tokenization method for code language processing.

- The evaluation on code generation benchmarks show that the tokenization can also improve language modeling performance in addition to inference efficiency.

**Weaknesses:**

Firstly, the details of the tokenization process are not entirely clear to me. For standard subword tokenization methods such as BPE or SentencePiece, these methods will not merge tokens that are separated by whitespace. For instance, in a sentence like "I go to the park every day.", these standard tokenization methods would not merge "go to" into a single token, nor would they merge "every day" into a single token, even if these phrases co-occur very frequently. I'm unsure if the method proposed in this paper follows the same principle as standard tokenization, refraining from merging tokens separated by whitespace. If this is the case, I have serious doubts about the claim that adding 1k vocabulary can outperform 1M vocabulary, as this seems unlikely from my understanding. I suspect that the method proposed in this paper might merge tokens separated by spaces, such as "go_to" or "every_day". If this is true, the comparison with the original BPE method seems unfair as they operate under different settings. Although I acknowledge that this approach may be more reasonable for code data, standard tokenization can also be easily applied to this setting. If so, the authors should provide more details and evaluations to demonstrate the superiority of their method.

Secondly, this method seems to be only applicable to code data, as code data contains many very frequent patterns (e.g., "int cnt = 0;". If we allow merging tokens beyond spaces, we can have "int_cnt_=_0;" as 1 token). This is why it makes sense to do so in code data. However, in the Natural Language (NL) domain, this approach would likely result in a significant reduction in performance improvement and could cause semantic confusion due to the tokenization. Therefore, I question the universality of this method and wonder if the authors have conducted experiments on natural language datasets to validate this method.

Thirdly, the improvement of this tokenization method on the code dataset is not substantial, with only about a 10% increase. Recent works have used neural methods for context compression, achieving a compression ratio of 2-4 times. The authors should discuss and compare their method with these techniques to demonstrate the value of FuseToken.

**Questions:**

See weakness

---

> ### Author Response · Authors · 2023-11-23
>
> We are very grateful for the time you spent carefully reviewing our work. You raise good points and look forward to incorporating them into future revisions.
>
> 1. All whitespace in all tokenizers presented in this work follows the standards set in other tokenizers by encoding every byte and replacing whitespace with special characters. As such all whitespace is treated as if it were any other character for the purpose of learning token encodings. Therefore both BPE and FusionToken have tokens that include both regular and whitespace characters.
> 2. With respect to NL, we acknowledge that code data, being fairly regular, can have great benefits from FusionToken. This however does not mean NL would be immune from benefits. Even with NL comprising a small percentage of the FusionToken dataset we still see a compression improvement of 2.57% (Table 1). With a greater percentage of text we would expect that number to be higher still.
> 3. We are not aware of any such neural compression paper that can improve compression by 2-4x as to our knowledge the state of the art in tokenization (TokenMonster) is only ~25% more compressive than baseline BPE (but at the cost of not being easily integrated into Language Model training)

---

### Official Review · Reviewer_z6Wp · 2023-11-01

**Soundness:** 3 good
**Presentation:** 3 good
**Contribution:** 2 fair
**Rating:** 5
**Confidence:** 2

**Summary:**

The paper introduces Fusion Token, a method that enhances the conventional Byte Pair Encoding (BPE) technique for data encoding in language models. Fusion Token utilizes a more aggressive computational strategy by expanding token groups from bi-grams to 10-grams. This approach results in a higher compression rate compared to regular BPE tokenization with a vocabulary of one million. The method demonstrates noticeable performance improvements and faster inference times, as it increases the data scope per compute unit and reduces the number of tokens per string. By dedicating more computational resources to the tokenizer building process, Fusion Token maximizes the efficiency and effectiveness of code language models as data compression engines.

**Strengths:**

The paper exhibits strong motivation and is easy to follow. In the pursuit of achieving the highest data compression ratio, it is justified to expand token groups from bi-grams to 10-grams. Although the method is relatively simple, I believe it should outperform the naive BPE model in practical scenarios. However, the observed performance gap in the experiments is not as significant as described. Grouping frequent n-grams together in a straightforward manner can indeed increase the data compression ratio, even without complex theories or networks behind it.

**Weaknesses:**

1. My main concern with this paper is the performance of token fusion on natural language (NL) corpora. The authors have only conducted experiments using a code corpus, which is significantly different from NL. Code language typically consists of similar patterns, better structure, and limited vocabulary size, making token fusion more likely to be effective. However, NL content is more complex, with greater variability in content. Therefore, it is essential for the authors to conduct experiments on an NL corpus to demonstrate the effectiveness of their method. In my opinion, token fusion should have a greater impact on code than on language.

2. The paper would benefit from conducting additional ablation studies in the experiments. I am particularly curious about the rationale behind selecting only 1K tokens in the paper. It would be interesting to explore the outcomes by varying the number of tokens, such as selecting 500 tokens, 2K tokens, or 4K tokens. This would provide valuable insights into how the performance changes with different token selections.

**Questions:**

1 How to decide the fusion token for different corpus? Why you choose 1K in your paper?
2 Does token fusion cross whitespace?

---

> ### Author Response · Authors · 2023-11-23
>
> Thank you very much for your careful reading of our work and insightful comments. We in fact considered the points you raise so thank you again for your suggestions.
>
> 1. While we agree the addition of NL tests would strengthen the work, we chose to focus on code because of its relative challenge in tokenization compression (as per Table 1 code compression is uniformly worse than NL compression) and for evaluation in a more robust fashion.
> 2. As a follow up to this work we intend to extend the work to NL and extend our experimentation further to answer these questions
> 3. Similarly, while the addition of ablations around fusion token length would assuredly strengthen the work, the primary objective of this work is to present the algorithm and compare it robustly against BPE. While information on how to best select the optimal fusion token size is important, it is less so and thus was left out of this submission.
> 4. Questions:
>     1. We use the union over all corpora to produce our fusion tokens
>     2. We chose 1K after performing ablations over the number of added tokens. These ablations however were not particularly notable and thus were not included in the full text
>     3. FusionTokens do cross whitespace, but in this case as whitespace are encoded into the byte space and replaced (as is the standard in modern tokenizers for LLMs) the BPE tokens can contain both whitespace and non-whitespace

---

### Official Review · Reviewer_6zo7 · 2023-11-01

**Soundness:** 2 fair
**Presentation:** 3 good
**Contribution:** 3 good
**Rating:** 3
**Confidence:** 4

**Summary:**

This paper presents a fusion token method. Given a dataset, the proposed method integrates the newly constructed n-gram tokens derived from that dataset into the existing BPE vocabulary. These newly added tokens are treated as special tokens, taking precedence over existing tokens during inference. Experimental results demonstrate that these added tokens enhance the compression rate of the dataset and achieve higher scores on code generation benchmarks.

**Strengths:**

The paper underscores the significance of optimizing tokenization by analyzing the relationship between bits per word and bits per byte. The proposed method is articulated as an algorithm, facilitating comprehension.

**Weaknesses:**

The main concerns regarding this paper are:

- A significant issue is the bias in the experimental setup. The existing BPE tokens are formed from data comprising multiple ``text’’s. However, the added tokens from Fusion token method become contingent upon the additional training data. Essentially, they hinge on $D$ provided as input in Algorithm 1. If the distribution of the training data and the data being evaluated are similar, the newly added tokens will notably influence the evaluation. In section 4.2, the training data was obtained from code written in various programming languages, with subsequent evaluations conducted on code benchmarks. Since both the dataset for building the additional tokens and the dataset for evaluation are similar, an enhanced performance over the conventional BPE is anticipated.

- The aforementioned bias is also observed in Table 1 and Figures 2(a) and 2(c). Tokens introduced via Fusion Token show a remarkable performance improvement in code part relative to text. Since the tokens were derived from code data, their impact in the experiments is profound.

- To validate the robustness of the tokenization, evaluations using a variety of benchmarks other than code generation are essential.

- Assessing the efficacy of the 1K tokens added by the proposed method necessitates a comparison against a tokenization with 1K tokens appended solely through bi-gram merging (existing BPE approach).

- Based on the data in Table 4, it's premature to infer that "BPE for large models with Fusion Token is superior to standard BPE" from tests on just two models.

**Questions:**

Q1: Of what are the actually added 1K tokens comprised?

---

> ### Author Response · Authors · 2023-11-23
>
> We very much appreciate your effort in careful review of our contributions and insightful points.
>
> 1. With respect to experimental setup, both the BPE and the fusion tokens are created from the same identical dataset. The Fusion tokens in fact use a subset of the dataset used to train the BPE tokens both being drawn from the code dataset. As a result there exists no marginal experimental bias between these two models.
> 2. As both the BPE tokens and FusionTokens are both trained on code data the impact of any such bias is non-existant and the improvements come only from the overall efficiency of FusionTokens not from any introduced bias
> 3. While we acknowledge that the addition of other, non-code generation experiments would strengthen the paper, given our models are trained primarily on code we opted to only evaluate on code generation tasks
> 4. We do indeed compare the 1K FusionTokens against BPE, in fact extending the BPE vocabulary up to 1M (all trained on the same exact code dataset that the 1K fusion tokens are drawn from) and show a substantial improvement over a BPE model with far larger vocabulary size
> 5. Questions:
>     1. The 1K tokens, like the BPE tokens, comprise ngrams (in BPE’s case bigrams) computed over the code dataset

---

### Official Review · Reviewer_oudB · 2023-11-06

**Soundness:** 1 poor
**Presentation:** 2 fair
**Contribution:** 2 fair
**Rating:** 3
**Confidence:** 3

**Summary:**

This paper present a Fusion Token method for tokenization, by expanding the token groups in BPE from bi-grams to 10-grams.
By adding extra 1024 10-grams tokens to  a 51K BPE vocabulary, this paper claims the compression rate surpasses that of a regular BPE tokenizer with a vocabulary of one million.

In experiments, this paper trains 125M LM and 650LM respectively, and shows the new vocabulary (51K + 1K) trained model is better than the 51K one for code generation task.

**Strengths:**

1. The proposed tokenization method show a faster inference time thanks to the shorter tokenized sequence length.

2. The proposed method brings improvement for code generation task.

**Weaknesses:**

Experiments and results alone are insufficient to support the claim. I believe this paper fails to demonstrate that the proposed method can enhance the performance of a language model.


- 1. Introducing an additional 1024 10-gram tokens can indeed result in a higher bytes-per-token value compared to vanilla BPE tokenization, which is an expected outcome and not surprising. The contribution of this higher compression rate is not clear, as the language model's performance does not appear to improve, as shown in Table 4.

- 2. The paper falls short in establishing connections between the results and existing language models, particularly in terms of data, model architecture, and downstream tasks. This paper trains two language models, one with 125 million parameters and another with 650 million parameters, using a subset of the Pile dataset, and evaluates them with two code generation tasks. There is a lack of discussion about the rationale behind this specific experimental setup, which makes it challenging to comprehend and verify the results since they cannot be directly compared to other papers. I have a question: Why not use widely recognized language model benchmarks such as PG-19 and WikiText103 for training and evaluation, or follow the setup described in the official Pile dataset paper to train GPT2 models of varying sizes (small, medium, large)?

- 3. Only the code generation downstream task is evaluated.

**Questions:**

How is "Bytes per token" in Table 1 calculated? Is it the average length E[l]?

It appears that the proposed method is not specifically designed for programming languages but rather for general language modeling. If that's the case, why was a programming language chosen for testing?

---

> ### Author Response · Authors · 2023-11-23
>
> Thank you very much for your comments. We appreciate the time you spent and for your thoughtful insights.
>
> Addressing Weaknesses and Questions:
> 1. The main contribution of the paper is the introduction of an algorithm that improves on the compression of BPE at the same vocabulary size without any harm in core language modeling tasks.
> 2. We do not train exclusively on the Pile dataset (See section 4.2) we train primarily on code data with only a relatively small percentage of text data. Code data is notably more difficult to achieve efficient compression (see Table 1) while at the same being easier to evaluate in an objective fashion.
> 3. It is for these reasons that our model, primarily trained on code, was only evaluated on code generation tasks.
> 4. Questions:
>     1. Bytes Per Token is calculated by tokenizing a dataset and calculating the number of bytes in all rows of the dataset, and the number of tokens for in total in the dataset and dividing bytes / tokens.
>     2. Programming languages are chosen due to their lower compression in traditional BPE based approaches and for the robustness in evaluation.

---

### Author Response · Authors · 2023-11-23

We want to thank the reviewers for their time and helpful comments. We would like to emphasize that the main benefit of the FusionToken paper is in the presentation and analysis of the addition of n-gram tokens to improve the compressive capabilities of tokenizers and language models without any adverse effect on generation performance. In fact we show that the use of FusionTokens results in improved performance on our code benchmarks. All tokens from FusionToken or BPE are created using the exact same dataset and distribution which is over code across multiple languages wherein all data is encoded to 256 byte representations and spaces are replaced with special characters denoting them. Our deliberate choice of code as the evaluation domain stems from its propensity for greater challenge when subjected to BPE alone, as evidenced by average compression results. This decision aligns with the established practice of employing code tasks as a more stringent benchmark than natural language problems, especially in the context of evaluation by large language models like GPT-4. Overall the usage of FusionTokens provides an improvement in compression, and thus inference speed without any measurable cost in model performance.

---

### Meta-Review · Area_Chair_TSKy · 2023-12-10

**Metareview:**

The paper proposes a method that expands on BPE for tokenization in language models by utilizing token groups from bi-grams up to 10-grams. It leads to improved compression rates and as a result faster inference times compared to standard BPE, while also improving performance on code generation tasks. The method is evaluated on code generation benchmarks.

Strengths:
- Most of the works on LLMs take the tokenization for granted and don't optimize it. This paper takes a drastic view by revising the baseline BPE  and demonstrates better compression performance.
- Tackling a very important subtask in language modeling: It shows improved language model performance on code generation with the proposed tokenization method.

Weaknesses:
- Evaluations were limited to code data and tasks, hence limiting the impact and applicability of the work by lacking demonstrations with natural language.
- The performance gains over standard BPE are relatively small.
- There potential weakness in the experimental setup since the tokens are derived from the same distribution. Hence, again, suggesting to better evaluate the effectiveness one needs to evaluate on benchmarks beyond code generation.

Suggested improvements for the next iteration of the paper:
- Evaluations on natural language data and other NLP tasks to demonstrate broader applicability.
- More comparisons  with standard BPE, for example limiting both to same vocabulary size.
- Experiments adding tokens derived from a different distribution than the training data to reduce potential bias.

**Justification For Why Not Higher Score:**

Limited applicability of the proposed method without proper comparison with the baseline BPE.

**Justification For Why Not Lower Score:**

N/A

---

### Decision · Program_Chairs · 2024-01-16

Reject